# Volumetric Changes in Morse Taper Connections After Implant Placement in Dense Bone. In-Vitro Study

**DOI:** 10.3390/ma13102306

**Published:** 2020-05-16

**Authors:** Georgios E. Romanos, Rafael Arcesio Delgado-Ruiz, Ana I. Nicolas-Silvente

**Affiliations:** 1Periodontology, School of Dental Medicine, Stony Brook University, Stony Brook, NY 11794-8712, USA; 2Prosthodontics and Digital Technology, School of Dental Medicine, Stony Brook University, Stony Brook, NY 11794-8712, USA; rafael.delgado-ruiz@stonybrookmedicine.edu; 3Restorative Dentistry, School of Dentistry, Murcia University, 30008 Murcia, Spain

**Keywords:** bone density, conical connection, deformation, dental implants, implant connection, implant index, volumetric changes

## Abstract

The stability of the implant–abutment interface is crucial for the maintenance of the implant index integrity. Several factors are capable of inducing deformation in implant–abutment connection, such as the actual insertion of the implant into the bone. This study aimed to evaluate the deformations produced in the connection after the insertion of the implant. Ten implants with two different implant carriers (Type A: carrier attached to platform and Type B: carrier attached inside the index-connection) were placed in artificial Type II bone, and volumetric changes were evaluated for different connection features with a 3D digital microscope. ANOVA (analysis of variance), Wilcoxon, and Tukey HSD post-test were used for statistical comparisons. Type A implants presented deformation at the platform level (inner slot angles and slot width), but no volumetric changes were observed inside the connection. Type B implants presented deformation in three parameters inside the connection (outer channel length, coronal step width, and coronal step length). Within the limitations of this study, we can conclude that more deformation is expected at the internal connection when the implant carrier engages this area. The engagement area should be as far away as possible from the index connection.

## 1. Introduction

One of the more frequent long-term complications in modern implantology is peri-implantitis. Recent studies showed a cumulative incidence of peri-implantitis among patients, between 24.4% and 39.7% [1,2]. In a recent meta-analysis, Derks et al. [3] estimated a mean prevalence of 22% (95% CI: 14–30%). The risk of peri-implantitis is significantly elevated in patients who smoke or have prosthetic mismatches [2]. A systematic review from Ferrera et al. [4] concluded that a history of periodontitis was also associated with the occurrence of peri-implantitis. However, it is crucial to determine the potential factors related to peri-implantitis for preventive strategies [4]. Some of the risk factors, which increase the incidence of peri-implantitis are the previous history of periodontitis, bacterial plaque, bleeding, bone level on the medium third of the implant, lack of prosthetic fit, or a non-optimal screw joint, among others [5]. Also certain systemic diseases, such as diabetes, have been related to a higher incidence of peri-implantitis [6]. The type of connection also seems to be essential, since various studies have shown a higher incidence of peri-implantitis in external connection implants compared to internal connection implants [7,8].

When using a two-piece dental implant system, the implant body and the prosthetic abutment are connected with a prosthetic screw. The implant–abutment interface must be stable to resist functional loads [9] and reduce the screw loosening [10]. To maintain this stability, especially in single-implant supported restorations, there is a mechanical structure called the implant/prosthetic index, which is located inside the implant’s body (internal connection) or outside the implant’s body (external connection). Depending on the implant system, this index has a defined geometry designed to prevent the rotation between the components. The maintenance of the integrity of the implant index is crucial. The type of the implant–abutment connection is also vital for the kind of bacterial colonization, being the bacterial species detected in the Morse-tapered connections are often associated with health [11,12].

There are several factors capable of inducing a deformation at the connection level. Those factors include overload, non-axial loads [13], frictional wear [14], the presence of micro-movements between the parts because of a misfit [15], a non-passive fit of the restoration [16], multiple connections and disconnections of the abutment [17], but also the insertion itself of the implant [18]. The forces generated by torque during the insertion process may cause morphological changes, from the deformation of the implant platform or the damage in the indexation to the implant breakage or bone fracture [19]. Between the different types of connection, the Morse-taper connection seems to be more stable because it causes less stress on the abutment screw [20]. However, Delgado et al. [21] showed that morphological changes occurred inside the internal connection when narrow implants were placed in dense bone; moreover, the deformation of the connection is accompanied with titanium particle release. This deformation suffered by the internal connection during the insertion process has been characterized by an enlargement of the micro-gap accompanied by increased wear and bacterial leakage [22], and for further development of a future peri-implant disease.

This deformation inside the connection is caused by the torsional forces of the implant carrier to overcome the resistance of the bone to the insertion of the implant [21]. To avoid this problem, a change in the way the implant is manufactured could be beneficial, protecting the connection morphology.

Therefore, the present study aimed to evaluate the deformations produced by two different implant carriers, one of them applied inside the internal conical connection, and the other one used over the implant platform after the insertion of the implant in artificial Type II bone.

## 2. Materials and Methods

### 2.1. Bone Samples

Artificial bone blocks (130 × 180 × 40 mm) made in polyurethane solid foam resembling Type II density bone (Sawbones, Pacific Research Laboratories, Vashon, WA, USA) were used. The compressive modulus, strength, and density of the bone blocks were 210 MPa, 8.4 MPa, and 20 pcf, respectively; also, the tensile strength was 5.6 MPa with a shear strength of 4.3 MPa.

### 2.2. Operator Calibration

The same calibrated single operator developed the drilling and insertion of all dental implants, following the manufacturers’ recommended drilling protocol at 800 rpm.

For the operator calibration, ten implant beds were created, and the implants were inserted ten times per group. The implants were placed with the platform at the bone level surface. The intraclass coefficient (ICC) was calculated to consider the operator calibration. A value equal or higher than 80% was considered as a reliable ICC; having 0 mm difference between the implant platform and the bone level surface was considered as 100% agreement; <0.5 mm difference between them was considered as 90% to 99% agreement; 0.5 to 1.0 mm difference was considered as 90% to 80% agreement, and 1.0 to 1.5 mm difference between both surfaces was considered as 80% to 70% agreement.

### 2.3. Sample Groups

Ten titanium grade 5 dental implants (C and C/X) with slight tapered macro-design and progressive threads were used (Ankylos^®^ dental implants, Dentsply-Sirona, Mannheim, Germany). The main difference between implants was the implant platform design and the mounting carrier (implant driver) as follows:

Type A (mounting carrier Type A):

Five implants (A3.5/11 mm) were characterized by a conical connection, without indexation and with four rectangular slots located over the implant platform to hold the mounting carrier during the implant insertion. Longitudinally, the connection geometry was made up of two halves (Figure 1).

Type B (mounting carrier Type B):

Five implants (B4.5/14 mm), with a conical connection characterized by a six channeled/lobed indexation (hexagon connection) with curved walls as anti-rotational features (Figure 1). The implant driver was placed inside the indexation part of the connection during the implant insertion. Longitudinally, the connection geometry was made up of three thirds.

For both implant diameters, the connection was identical. The only difference was the implant length and diameter, as well as the type of mounting carrier (driver) connected with the implant. Specifically, the driver was connected with the implant platform on the rectangular slots (C-implants, Type A) or within the implant body using the six-channel index (C/X-implants, Type B). The entire implant–abutment connection was a Morse-tapered connection with a 6-degree angle.

### 2.4. Implant Insertion and Removal Procedure

The operator followed the manufacturers’ recommended drilling protocol for dense bone Type II, with an 800 rpm speed and without irrigation, as follows:

The first drill used was the pilot drill (2 mm diameter), then the twist drill (3 mm diameter) for Type A implants (narrow-diameter implants), and then (for the wider Type B implants with 4.5 mm diameter) the twist drill for Type B implants (4 mm diameter) was used to drill the bone with 800 rpm. The osteotomies were performed in the representative length under copious irrigation using a saline solution as external cooling system in the Frios Unit S/I with a Frios contrangle handpiece (Dentsply, Sirona, Mannheim, Germany). The osteotomy was enlarged in the crestal part with the conical reamer (A- or B-size, dependent on the implant diameter with corresponding diameters 3.5 and 4.5 mm, respectively) and the tapping of the osteotomy was performed manually. Implants were inserted by hand using a ratchet.

The insertion torques were recorded when the implants were inserted leveled with the bone surface (yuxtacrestal) according to the manufacturer’s protocol, and the insertion torque of the implant was recorded.

The implants were removed from the bone block with a ratchet and were immersed in an ultrasonic bath with distilled water for 30 min to remove remnants of artificial bone from the surface.

Then, the implant carriers were detached from the implant bodies for further evaluation of the implant–abutment interface.

### 2.5. Evaluation of the Volumetric Changes of the Implant Connection

To evaluate the volumetric changes that occur in the Morse taper implant–abutment connection after its placement and removal in Type II bone, a last generation digital microscope (Keyence VHX-6000, Keyence Corporation, Osaka, Japan) was utilized. A vertical scanning step of 10 microns, with high-resolution mode (HDR), with 50x magnification, was completed. The objective was focused on the deepest portion of the connection, and the scanning process started at the bottom, followed a down-up direction until the most superior aspect of the platform was fully scanned. A 3D-reconstruction file of the scanned 4–6 mm coronal segment was obtained. The measurement software from the same manufacturer was used (SK-H Data Acquisition Software, version 1.0.3.0. (2014), Keyence Corporation, Osaka, Japan).

According to the 3D reconstruction of the implant connection, different linear and angular measurements were made to evaluate the connection volumetric changes, comparing the measures before the implant insertion, after the implant insertion, and after removal of all the implants. The following measurements were evaluated:

For Type A implants (Figure 2):-Implant diameter: 3.5 mm: Four different diameters were measured in each implant: (a) external edge of the platform; (b) coronal step diameter of the connection; (c) middle step diameter of the connection; and (d) apical step diameter of the connection.-Antirotational slot angles: Each platform had four antirotational rectangular slots on the surface, characterized with four angles each: two outer angles, near the implant external edges; and two inner angles, near the implant connection walls. Measurements were performed in both outer and inner slot angles.-Antirotational slot length and width: The two lengths and two widths of each slot rectangle were measured.

To evaluate the changes of the implant connection walls, a sagittal section of the 3D-overview of the connection was made evaluating the different parts of its components. A profile of the connection was generated through the software, and both horizontal and vertical measurements were made in each of the two segments of the connection as follows:-Profile horizontal measurement: The step between the upper half and the lower half of the connection was measured, and it was called the middle step width. This measure allowed us to evaluate narrowing or widening in the connection diameter.-Profile vertical measurement: The length of the upper half (coronal step length) and the lower half (apical step length) of the connection were measured. With this parameter, we evaluated possible shortening or lengthening in the connection.

For Type B implants (Figure 3):-Implant diameter: 4.5 mm: the more coronal outer diameter was measured.-Volumetric changes of the antirotational feature, internal hexagon—vertex angle of the hexagonal connection: the inner angles formed between the straight sides of the internal hexagon connection were obtained. Six angles were obtained by each implant.-Volumetric changes of the antirotational feature, internal hexagon—length of the hexagonal connection: each vertex was connected with the nearest one obtaining six straight lines, which represented the length of each hexagon side. Six side lengths were obtained and measured per implant.-Channeled indexation measurements: Six channeled indexation rectangles with curved walls as anti-rotational features were located inside the connection. The following features were measured:(a)Outer channel length: Longitudinal measurement of the outer longest side of each rectangle channel was measured.(b)Inner channel length: Longitudinal measurement of the inner longest side of each rectangle channel was measured. As there were six rectangle channels per implant, six outer and six inner measures were taken per implant.(c)Channel width: Longitudinal measurement of the lateral side of the rectangle channel. The two shorter sides of each rectangle channel were measured, and it was described as “channel width”. As there were six rectangle channels per implant, twelve measures were taken.(d)Channel index inner angle: Angular measurement of the two internal angles of each channel was evaluated. Twelve inner angles per implant were measured.-Wall thickness: The width of the connection “outer wall” was measured at the beginning of the coronal third (platform wall width or outer wall) and also was measured at the end of the coronal third (apical wall width or inner wall). The mean between both measurements was considered the mean wall width.

Again, a sagittal section of the 3D-overview of the connection was made to evaluate both horizontal and vertical measurements, as follows:-Profile horizontal measurement: The step between the coronal third and the middle third and the step between the middle third and the apical third of the connection were measured.-Profile vertical measurement: The lengths of the coronal, middle, and apical third of the connection were measured.

### 2.6. Statistical Analysis

The volumetric changes suffered by the connections were evaluated with an analysis of variance (ANOVA) for all the variables using the SPPS version 23.0 statistical package (SPPS Inc., Chicago, IL, USA).

For group A, the Wilcoxon test was used for the statistical comparison before and after the implant insertion. For group B, the Tukey HSD test was used for the statistical comparison before and after the implant insertion, both with a statistical significance set at *p* < 0.05.

## 3. Results

The insertion torque was controlled for each implant placed and was in a range between 50 and 60 Ncm applying homogeneous stress to all the connections.

All the parameters were measured before and after placing the implants in bone with Type II density. All implants showed visible wear and titanium delamination in some parts of the platform or connection, depending on the implant site in contact with the carrier.

For Type A implants, volumetric changes were observed after the implant insertion, expressed as deformation of the inner slot angles, and increased slot width as follows:-Inner slot angles with a mean angle of 84.95° (before) and 85.35° (after) (*p* = 0.049, Wilcoxon test).-Slot width with a mean width of 534.21 µm (before) and 546.95 µm (after) (*p* = 0.011, Wilcoxon test).

However, at the Type A implant connection walls, no changes were observed, expressed as stable measurements in the remaining variables of analysis (Table 1).

Both parameters altered after insertion in dense bone were located in the slots, at the platform level, none of the parameters evaluated were affected inside the connection (Figure 4).

When evaluating the Type B implants statistics showed significant differences in the following parameters:-Outer channel length, with a mean length of 542.87 µm (before) and 532.59 µm (after) (*p* = 0.001, Tukey HSD test).-Coronal step width, with a mean length of 287.67 µm (before) and 329.25 µm (after) (*p* = 0.031, Tukey HSD test).-Coronal step length, with a mean length of 2418.96 µm (before) and 2484.99 µm (after) (*p* = 0.009, Tukey HSD test).

The other parameters evaluated in implant Type B did not show statistical differences and are presented in Table 2. The three parameters altered are located inside the connection, more precisely in the coronal part of the connection, with two of them in the indexation part. The grade of deformation in the Type B model is shown in Figure 5.

## 4. Discussion

In our study, we found that a 50–60 Ncm of torque insertion resulted in morphological changes of the implant–abutment interface. The drilling protocol was strictly carried out by an experienced clinician, following the manufacturer’s instructions. The magnitude of the volumetric changes were evaluated; the measurements were performed using a last-generation digital microscope, which allows a 3D-evaluation of the implant index without destroying or distorting the samples.

The pattern of titanium particles, and delamination observed after the implant insertion in both groups, confirms the results obtained by Delgado-Ruiz et al. [21], which describe the process of implant insertion as a potential source of titanium particles and ions. Recent systematic reviews have already mentioned the release of titanium particles caused by the implant insertion itself [23,24]. These titanium particles could be generated from different parts of the system: the implant surface that is torn when rubbed into the bone, the inside of the connection that is damaged during the insertion process, or even the implant driver.

Several studies have evaluated the changes produced in the implant–abutment connection after the application of different values of insertion torque; some of them found differences from only 45 Ncm [25], while other authors needed higher values of insertion torque to see morphological changes in the connection, reaching values of 120 and 269 Ncm [19,26]. In the present work, we found some volumetric deformations with lower insertion torque values on implants with Morse-tapered connections. Luckily, there is no deformation in the inner conus but only in the anti-rotational index. Most of the studies at the moment have described the type of distortions, but there is a lack of information about the quantity of deformation. Few authors have evaluated the amount of strain in each component of the connection.

When evaluating the results for Type B implants, no morphological changes were observed in the length or width of the walls. This is in agreement with the results described by Delgado et al. [21], concluding that the forces applied during the insertion were transferred directly to the prosthetic index, but not to the external walls. All damages due to friction in Type B implants were located in the coronal part of the connection, and more precisely, in the anti-rotational index. This modification in the index morphology is going to lead to a mismatch with the prosthetic–abutment, increasing due to occlusal wear, which may lead to micromovements, especially when the implants are non-splinted. This complication has been reported in the literature for different butt–joint implant–abutment connections. The wear and the bacterial leakage increase in the presence of bacteria in the connection as well as inflammatory cells at the surrounding soft tissues, facts that are related, as already known, with the development of a future peri-implantitis [27,28]. In contrast to these connections, due to the effect that the conus geometry is not modified, there is a lack of micromovement and, therefore, a bacterial sealing under in vivo loading conditions [11], but there is a risk of fracture of the internal screw.

Scarano et al. [29] also evaluated the sealing capability of the implant healing screw in different types of internal connections, hypothesizing that the dimension of the micro gap between the implant and the screw affects the soft tissue healing. They found that internal hexagon connections were associated with a higher rate of volatile organic compounds (VOCs) emanating from the inner part of the implants compared with Morse-cone connection. The misfit in the micro gap interface can allow passage of fluids and bacteria, which is shown clinically as bleeding and malodor due to the presence of anaerobic bacteria. When evaluating the internal hexagon connection with 3D X-ray microtomography, numerous gaps and voids are present between the implant body and the abutments, increasing the mechanical stress on connection structures and surrounding bone tissue. In contrast, in the Morse-cone connection, no detectable separations between both components are found showing an absolute congruity, proving to be more tight and stable from a biomechanical point of view than flat-to-flat connections [30].

In implant Type A, morphological alterations were located over the implant platform, where the implant carrier attaches to the implant. The inner part of the implant did not show any morphological change.

This damaged area of the implant is not intimately in touch with the prosthetic abutment. Due to the implant feature using platform switching, subcrestal implant placement, and the one-abutment concept, the Morse taper implant–abutment connection may warrant long-term crestal bone stability, thanks to the mechanical properties of the conical connections [31].

The present study has been carried out on artificial bone. This may present some disadvantages since the results and conclusions can not be extrapolated to an in vivo situation. Future in vivo studies should be performed to evaluate the effects on animal or human models. On the other hand, the use of artificial bone also has some advantages, such as the homogeneity of the cancellous structure, and reproducibility of the method.

Another limitation of the present study could be the different diameter used in both experimental groups. Unfortunately, this kind of implant with the slots ubicated in the platform to connect the implant carrier is no longer commercialized, so we had a limitation in choosing the diameter since we only had this possibility available in stock. In order to overcome this limitation, the wall thickness of the implant was analyzed.

The wall thickness is usually the main difference in implants with different diameters, and not so much the parameters located inside the index, which are the same regardless of the implant diameter. The diameters were also measured in all portions of the implant (platform, cervical third, middle, and apical third), to evaluate if there was any type of deformation or elongation after the insertion into hard bone. None of these parameters were altered, neither the thickness of the walls nor the diameters, so we consider that the difference in diameter in both experimental groups does not imply such a limitation as we might initially suppose. Another limitation to be discussed could be the small sample, consisting of five implants per group. Still, it should be noted that the sample size for most of the parameters was not 5 (number of implants) since the measured element appears several times in the geometry. So, for instance, in “outer channel length”, where six channels appear for each implant, n was equal to 30 (6 × 5); in channel angle, as there were 12 angles per implant, n = 60 (12 × 5), and so on, being the sample adequated to draw reliable conclusions statistically [32].

The strength of the present work lies in the exact experimental protocol utilized with the use of a high-precision, non-destructive technology for the evaluation of the geometry and deformation suffered by the samples.

The clinical relevance of the present study is that the deformations at the Morse taper implant–abutment connection during implant insertion can increase the micro-gap, the micromovements, and potentially will increase the risk of mechanical and biological failures. The use of implant carriers engaging areas far away from the implant connection could reduce the risk of these failures in two-piece dental implant systems. The Morse-tapered connection may provide structural integrity if the antirotational features are not incorporated within the implant body; thus, avoiding wear under loading conditions.

## 5. Conclusions

-The implant insertion with carriers with the anti-rotational index at the implant platform seems to be beneficial since they displace the deformation far away from the walls of implant connection.-More deformation is expected at the Morse taper implant–abutment connection when the implant carrier engages this area.-Manufacturing changes may promote innovations to provide more stable and accurate implant carriers and Morse taper implant–abutment connection systems.

## Figures and Tables

**Figure 1 materials-13-02306-f001:**
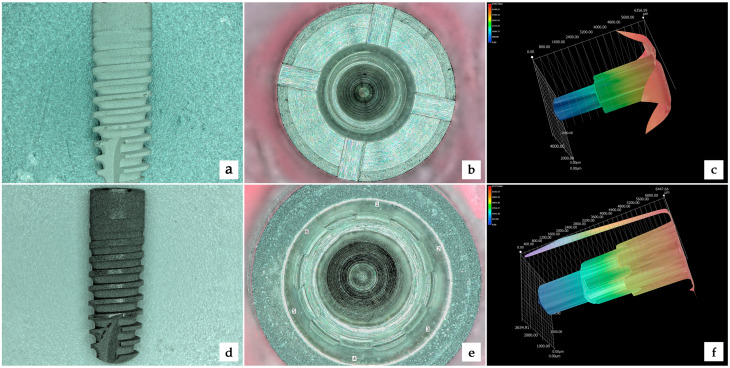
Implant features: (**a**) Type A—implant external overview; (**b**): Type A—implant platform overview; (**c**): Type A—D-scanned internal connection overview; (**d**): Type B—implant external overview; (**e**): Type B—platform overview; and (**f**): Type B—3D-scanned internal connection overview.

**Figure 2 materials-13-02306-f002:**
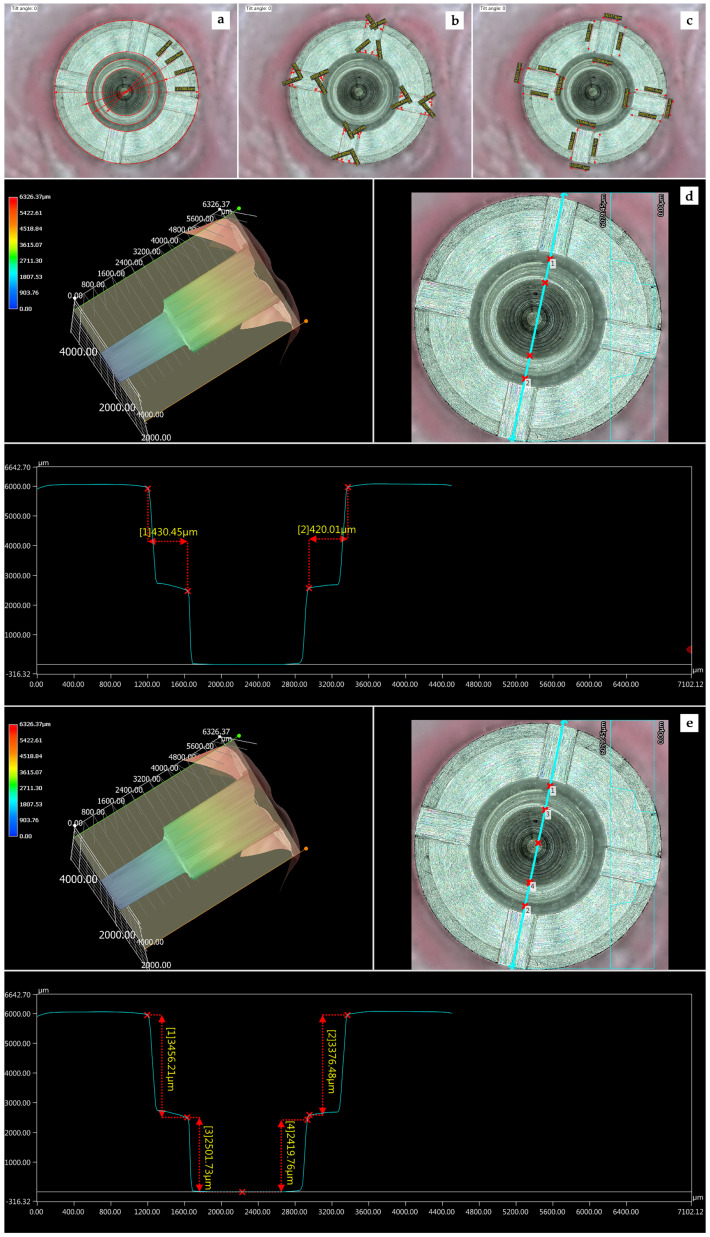
Parameters evaluated in Type A implants: (**a**) four diameters: external edge, coronal step connection, middle step connection, apical step connection; (**b**) outer and inner slot angles; (**c**) slot length and width; (**d**) profile horizontal measurements; and (**e**) profile vertical measurements.

**Figure 3 materials-13-02306-f003:**
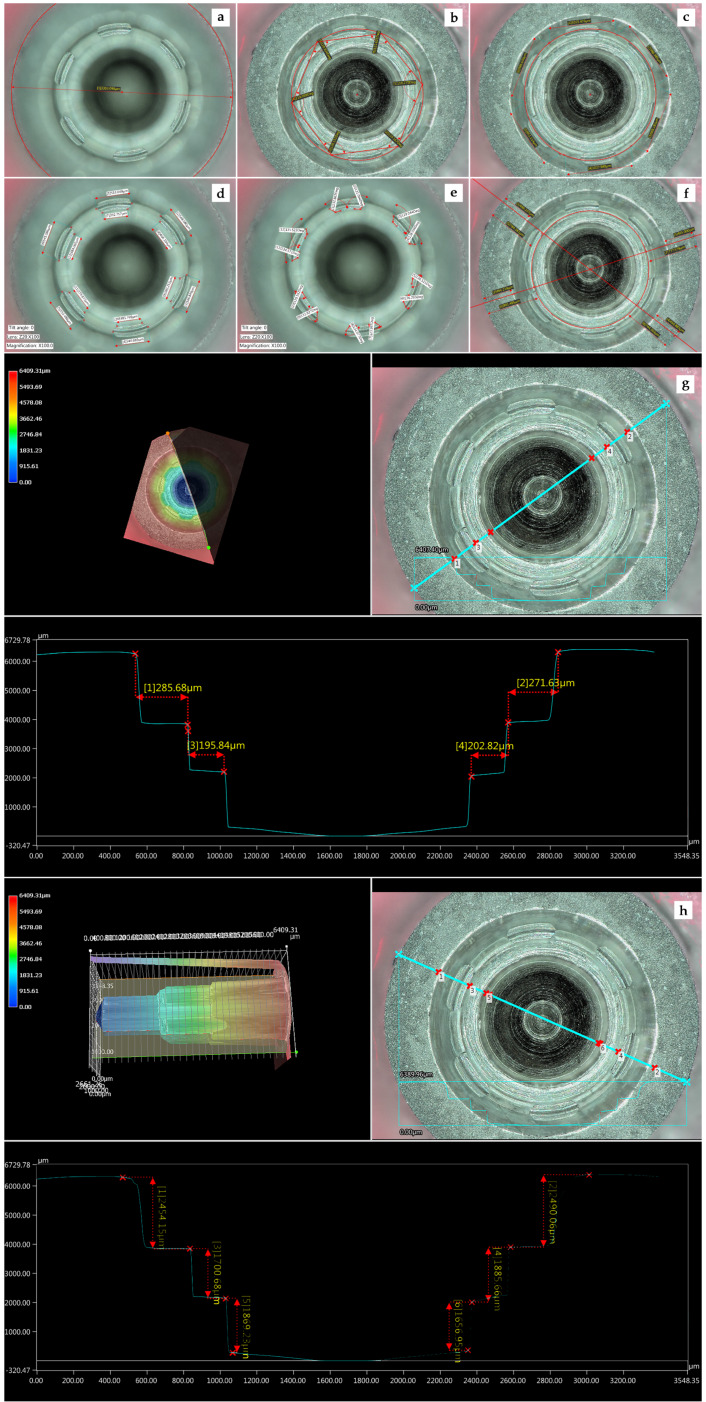
Parameters evaluated in Type B implants: (**a**) outer diameter; (**b**) vertex angle of the hexagonal connection; (**c**) length of each side of the hexagonal connection; (**d**) outer and inner channel length; (**e**) channel index angle; (**f**) wall dimensions; (**g**) profile horizontal measurements; and (**h**) profile vertical measurements.

**Figure 4 materials-13-02306-f004:**
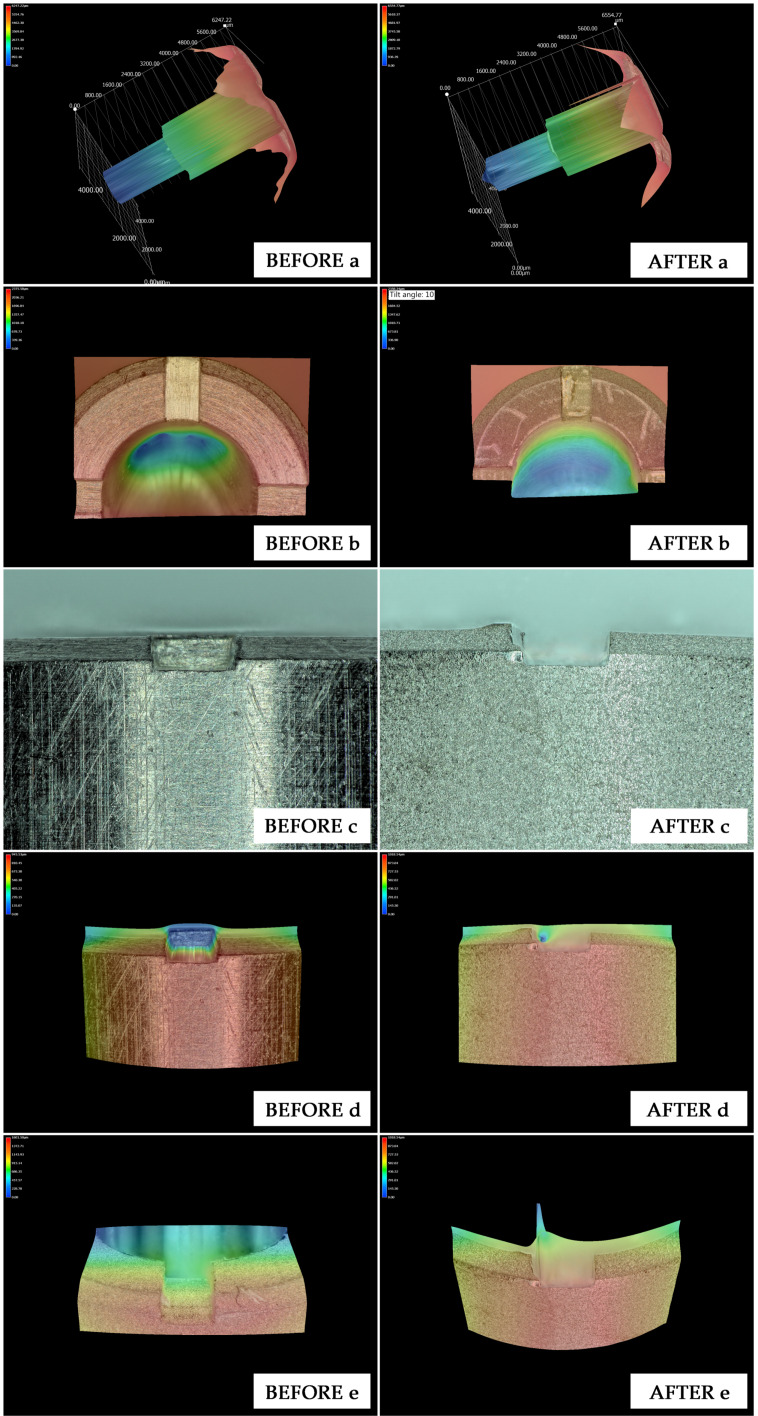
Comparison before and after placing the implant with carrier Type A in artificial bone Type II. (**a**) scanned 3D-reconstruction of the internal connection overview; (**b**) slots located in the platform (3D-reconstruction of the zenith view); (**c**) slots located in the platform (lateral view); (**d**) slots located in the platform (3D-reconstruction of the lateral view); (**e**) slots located in the platform (60° Tilted 3D-reconstruction).

**Figure 5 materials-13-02306-f005:**
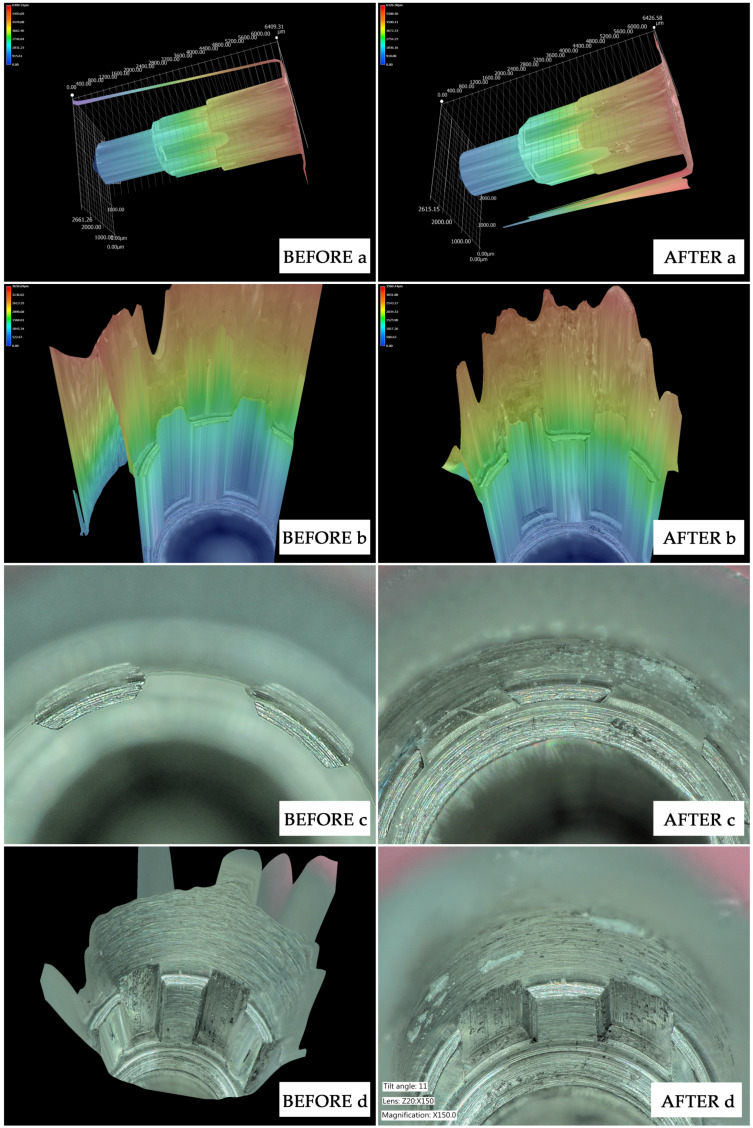
Comparison before and after placing the implant with carrier Type B in artificial bone Type II. (**a**): scanned 3D-reconstruction of the internal connection overview; (**b**): channels located in the index (3D-reconstruction); (**c**): channels located in the index; (**d**): channels located in the index (60° tilted 3D reconstruction).

**Table 1 materials-13-02306-t001:** Measurements (means) of the parameters evaluated for carrier Type A before and after insertion in density Type II bone. α and β mean the presence of statistical differences (Wilcoxon test, *p* < 0.05). Measurements with * means the presence of significative differences.

Type A	Unit	Before (Mean)	After (Mean)	*p*-Value
Diameters				
External Edge	µm	4384.10	4383.22	=0.925
Coronal Step Diameter	µm	2443.80	2428.06	=0.056
Middle Step Diameter	µm	2014.10	2011.32	=0.876
Apical Step Diameter	µm	1413.25	1434.56	=0.841
Slot Angles				
Outer Slot Angles	°	95.36	95.66	=0.400
Inner Slot Angles *	°	84.95 α	85.35 α	=0.049
Slot Dimensions				
Slot Length	µm	975.21	962.90	=0.092
Slot Width *	µm	534.21 β	546.95 β	=0.011
Profile Horizontal Measurements				
Middle Step Width	µm	432.90	457.83	=0.527
Profile Vertical Measurements				
Coronal Step Length	µm	3428.30	3410.29	=0.784
Apical Step Length	µm	2521.62	2638.53	=0.102

**Table 2 materials-13-02306-t002:** Measurements (means) of the parameters evaluated for carrier Type B before and after insertion in density Type II bone. α, β, and δ mean the presence of statistical differences (Tukey HSD test, *p* < 0.05). Measurements with * means the presence of significative differences.

Type B	Unit	Before (Mean)	After (Mean)	*p*-Value
Diameter				
Outer Diameter	µm	3344.69	3421.58	=0.108
Vertex Angle				
Connection Angle	°	119.99	120.03	=0.847
Side Length				
Connection Side Length	µm	1005.53	1005.78	= 0.965
Channel Index Measurements				
Outer Channel Length *	µm	542.87 α	532.59 α	= 0.001
Inner Channel Length	µm	383.57	378.99	= 0.232
Channel Width	µm	102.21	105.49	= 0.134
Inner Channel Angle	°	130.01	128.88	= 0.346
Wall Thickness				
Outer Wall	µm	455.74	458.84	= 0.709
Inner Wall	µm	790.50	787.57	= 0.790
Mean Wall	µm	623.12	623.21	= 0.989
Profile Horizontal Measurements				
Coronal Step Width *	µm	287.67 β	329.25 β	= 0.031
Middle Step Width	µm	209.18	201.58	= 0.609
Profile Horizontal Measurements				
Coronal Step Length *	µm	2418.96 δ	2484.99 δ	= 0.009
Middle Step Length	µm	1709.37	1698.22	= 0.133
Apical Step Length	µm	1828.64	1758.89	= 0.214

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
