# Peer review of "Volumetric Changes in Morse Taper Connections After Implant Placement in Dense Bone. In-Vitro Study"

_materials, 2020, doi:10.3390/ma13102306_

Round 1
Reviewer 1 Report
ABSTRACT:
page 1, lines 14-15: I would suggest to briefly explain the conical connections (type A and B).
INTRODUCTION:
page 1-2, lines 32-51: I would suggest to stress the difference between internal and external connection.
In line 45, the implant/prosthetic index is defined as a mechanical structure. Is it correct?
page 2, line 51: please add some updated systematic reviews to this sentence; as a suggestion, please read the article below:
Patini R, Staderini E, Lajolo C, Lopetuso L, Mohammed H, Rimondini L, Rocchetti V, Franceschi F, Cordaro M, Gallenzi P. Relationship between oral microbiota and periodontal disease: a systematic review. Eur Rev Med Pharmacol Sci. 2018 Sep;22(18):5775-5788.
page 2, line 57: do you have a synonym of "ruptur"?
METHODS:
Did you perform a sample size calculation?
page 3, lines 84-90There are several types of ICC. As a suggestion, you can read the paper below and apply their fomulas:
Koo TK, Li MY. A Guideline of Selecting and Reporting Intraclass Correlation Coefficients for Reliability Research. J Chiropr Med. 2016 Jun;15(2):155-63. doi: 10.1016/j.jcm.2016.02.012. Epub 2016 Mar 31.
I would move the information regarding torque insertion of the implant from the discussion (page 11, lines 277-278) to Matherials and Methods.
DISCUSSION:
page 12, line 299: I think that the adjective "significant" could be misleading, as the sample size is relatively small.
The paragraph with limitations and strenghts of the study is missing.
I think that this issue could bias the interpretation of the results.
Author Response
Dear Reviewer,
Thank you for your time and comments that certainly improve the quality and clarity of our work.
Following your recommendations, the following modifications were completed:
- ABSTRACT: page 1, lines 14-15: I would suggest to briefly explain the conical connections (type A and B).
Differences between the two types have been explained.
- INTRODUCTION: page 1-2, lines 32-51: I would suggest to stress the difference between internal and external connection.
Differences between both connections have been added in lines 42-44, with new bibliography [7,8]
- In line 45, the implant/prosthetic index is defined as a mechanical structure. Is it correct?
Correct.
- page 2, line 51: please add some updated systematic reviews to this sentence; as a suggestion, please read the article below:
Patini R, Staderini E, Lajolo C, Lopetuso L, Mohammed H, Rimondini L, Rocchetti V, Franceschi F, Cordaro M, Gallenzi P. Relationship between oral microbiota and periodontal disease: a systematic review. Eur Rev Med Pharmacol Sci. 2018 Sep;22(18):5775-5788.
The recommended bibliography has been added to this paragraph.
- page 2, line 57: do you have a synonym of "ruptur"?
"rupture" has been changed for "breaking"
- METHODS: Did you perform a sample size calculation?
No, we didn't. Only five implants for each group were used because, as added in a new paragraph in the discussion (lines 326-342): "One of the limitations of the present study could be the different diameter …..", but it should be noted that the sample size for most of the parameters was not 5 (number of implants), since the measured element appears several times in the geometry. So, for example, in "outer channel length", where six channels appear for each implant, n was equal to 30 (6*5); in channel angle, as there were 12 angles per implant, n=60 (12*5), and so on.
- I would move the information regarding torque insertion of the implant from the discussion (page 11, lines 277-278) to Matherials and Methods.
We decided to “discuss” the torque in this section because the maximum torque was not set up previously, so it was not defined in materials and methods. We did not limit the maximum torque. The maximum torque acquired was indeed a “result”. The drilling sequence was followed for each implant system for type II bone as recommended by the manufacturer. We recorded the maximum torque required to insert the implant in the ideal yuxtacrestal position, and all data registered were in a range between 50-60 Ncm. That’s the reason because we treat this issue in discussion and not in materials and methods.
- DISCUSSION: page 12, line 299: I think that the adjective "significant" could be misleading, as the sample size is relatively small.
Following your recommendation, the term “significant” has been deleted.
- The paragraph with limitations and strenghts of the study is missing.
New paragraphs with limitations and strength has been added (lines 326 – 345).

Reviewer 2 Report
INTRODUTION
Line 40-41 – And diabetes?
For example:
Naujokat H, Kunzendorf B, Wiltfang J. Dental implants and diabetes mellitus-a systematic review. Int J Implant Dent. 2016;2(1):5. doi:10.1186/s40729-016-0038-2
MATERIALS AND METHODS
Line 99 and 108 – How did the authors find the n of the groups?
n = 5 seems to me to be too small to infer statistically.
DISCUSSION
The authors do not discuss the limitations of the study, such as the reduced number of samples and the fact that they chose artificial bone
Author Response
Dear Reviewer,
Thank you for your time and comments that certainly improve the quality and clarity of our work.
Following your recommendations, the following modifications were completed:
INTRODUTION
Line 40-41 – And diabetes?
For example:
Naujokat H, Kunzendorf B, Wiltfang J. Dental implants and diabetes mellitus-a systematic review. Int J Implant Dent. 2016;2(1):5. doi:10.1186/s40729-016-0038-2
The recommended bibliography has been added: [6]
MATERIALS AND METHODS
Line 99 and 108 – How did the authors find the n of the groups?
n = 5 seems to me to be too small to infer statistically.
Only five implants for each group were used because, as added in a new paragraph in the discussion (lines 326-342: “One of the limitations of the present study could be the different diameter …..”, but it should be noted that the sample size for most of the parameters was not 5 (number of implants), since the measured element appears several times in the geometry. So, for example, in “outer channel length”, where six channels appear for each implant, n was equal to 30 (6*5); in channel angle, as there were 12 angles per implant, n=60 (12*5), and so on.
DISCUSSION
The authors do not discuss the limitations of the study, such as the reduced number of samples and the fact that they chose artificial bone
In the new paragraph (lines 326-342) mentioned above we discuss the limitations of the sample

Reviewer 3 Report
This study assessed the deformations produced in the connection after the insertion of ten titanium dental implants with two different implant platform design in artificial type II bone. The manuscript includes five figures with twenty-eight different sections, two tables, and twenty-five references. In general, it is a correct article, although some suggestions are made.
Please consider adding the word “implant” to the title as follows “Volumetric Changes in Conical Connections After Implant Placement in Dense Bone. In-Vitro Study”. In this way, I think the title would be more explanatory.
The manuscript contains six keywords. For keywords, where possible, please use Medical Subject Headings Terms (MeSH Terms). An alternative MeSH term proposed is “bone density” better than “dense bone”. The following terms “Conical Connection”, “Deformation”, “Implant Index”, “Implant Connection”, and “Volumetric Changes” are not MeSH terms.
Page 1, lines 34 or 36, and page 2, line 60. Please, place the reference number immediately after the article's first author name as you do on page 11, line 284.
Please, besides the statistical tests, indicate the statistical program used.
In tables, instead of expressing p-value with an inequality statement (p>0.05), use the actual p-value as, e.g., p=0.35.
Although the references are presented in the correct format according to the journal’s guidelines; on some references, the volume does not appear in italics. Please correct this where necessary.
Author Response
Dear Reviewer,
Thank you for your time and comments that certainly improve the quality and clarity of our work.
Following your recommendations, the following modifications were completed:
- Please consider adding the word “implant” to the title as follows “Volumetric Changes in Conical Connections After Implant Placement in Dense Bone. In-Vitro Study”. In this way, I think the title would be more explanatory.
We agree that this change is appropriate and we proceed to change the title: “Volumetric Changes in Conical Connections After Implant Placement in Dense Bone. In-Vitro Study”
- The manuscript contains six keywords. For keywords, where possible, please use Medical Subject Headings Terms (MeSH Terms). An alternative MeSH term proposed is “bone density” better than “dense bone”. The following terms “Conical Connection”, “Deformation”, “Implant Index”, “Implant Connection”, and “Volumetric Changes” are not MeSH terms.
We recognize that it is advisable to use Keywords indexed in MeSH. Unfortunately, there are not too many indexed words that are specific for such a particular article. Following the “Materials” recommendations for authors, we have tried to insert ”keywords specific to the article.”
- Page 1, lines 34 or 36, and page 2, line 60. Please, place the reference number immediately after the article's first author name as you do on page 11, line 284.
The reference has been added as recommended.
- Please, besides the statistical tests, indicate the statistical program used.
The paragraph: “The volumetric changes suffered by the connections were evaluated with an analysis of variance (ANOVA) for all the variables using the SPPS version 23.0 statistical package (SPPS Inc., Chicago, IL, USA).” has been added to Materials & Methods.
- In tables, instead of expressing p-value with an inequality statement (p>0.05), use the actual p-value as, e.g., p=0.35.
The actual p-value has been updated and expressed in both tables.
- Although the references are presented in the correct format according to the journal’s guidelines; on some references, the volume does not appear in italics. Please correct this where necessary.
Thank you for pointing out that detail. “Volume” changed to italics.

Reviewer 4 Report
The study assesses the deformation due to the insertion and removal of two implants with a different platform design. The first platform has an internal index, and the second platform has the index located coronally.
The study appears well done, the purpose is clear, the materials and methods are appropriate, the results reported correctly, the discussion is clear and centered on the topic.
Some doubts remain:
- What are the implants made of? The mechanical characteristics are influenced by the construction material of the implant. CP titanium or alloy?
- Why did the implants used in the two groups have two different diameters? Could the thickness of the titanium at the connection level have influenced the results? Please discuss the topic in the discussion
- Line 130: “The insertion torques were recorded when the implants were inserted leveled with the bone surface and were in a range between 50 and 60 Ncm.” The relationship between the insertion torque and the degree of deformation may be interesting. In addition, 50 Ncm is a high torque for implant placement. Why was this torque choosen? Please discuss it.
- The figures are too small and do not allow you to appreciate the connection design.
Author Response
Dear Reviewer,
Thank you for your time and comments that certainly improve the quality and clarity of our work.
Following your recommendations, the following modifications were completed:
1. What are the implants made of? The mechanical characteristics are influenced by the construction material of the implant. CP titanium or alloy?
Titanium grade 5. It has been added to material & Methods, line 96.
2. Why did the implants used in the two groups have two different diameters? Could the thickness of the titanium at the connection level have influenced the results? Please discuss the topic in the discusión
The following new paragraph has been added to the discussion to clarify this point:
“One of the limitations of the present study could be the different diameter used in both experimental groups. Unfortunately, this kind of implant with the slots ubicated in the platform to connect the implant carrier is no longer commercialized, so we had a limitation in choosing the diameter since we only had this possibility available in stock. Precisely because of this difference in diameter, one of the parameters evaluated was the wall thickness. The wall thickness is usually the main difference in implants with different diameters, and not so much the parameters located inside the index, which are the same regardless of the implant diameter. The diameters were also measured in all portions of the implant (platform, cervical third, middle, and apical third), to evaluate if there was any type of deformation or elongation after the insertion into hard bone. None of these parameters were altered, neither the thickness of the walls nor the diameters, so we consider that the difference in diameter in both experimental groups does not imply such a limitation as we might initially suppose.”
3. Line 130: “The insertion torques were recorded when the implants were inserted leveled with the bone surface and were in a range between 50 and 60 Ncm.” The relationship between the insertion torque and the degree of deformation may be interesting. In addition, 50 Ncm is a high torque for implant placement. Why was this torque choosen? Please discuss it.
We did not limit the maximum torque. The drilling sequence was followed for each implant system for type II bone as recommended by the manufacturer. We recorded the maximum torque required to insert the implant in the ideal yuxtacrestal position. All data registered were in a range between 50-60 Ncm.
4. The figures are too small and do not allow you to appreciate the connection design.
We have all the high-resolution images available to provide them to the editor if he deems it appropriated.

Round 2
Reviewer 1 Report
TITLE: I would suggest "Volumetric changes of different implant-abutment interfaces (rectangular slots vs hexagon connection) after placement in dense bone. In-Vitro Study" or "Volumetric changes of different implant-abutment interfaces (internal vs external connection) after placement in dense bone. In-Vitro Study"
ABSTRACT:
page 1, lines 14-16: I would rephrase this sentence, e.g.: "The stability of the implant-abutment interface is crucial the maintenance of the implant integrity index". What is implant integrity index? Do you mean clinical long-term success?
page 1, line 17: please change the term "structure" with "implant-abutment connection" and try to use this term for the whole article. As well, sometimes you use "conical connection" and sometomes "Morse taper connection"; the use of many synonyms is measliding for the readership. Please try to use the same term for the whole article "Morse taper implant-abutment connection".
page 1, line 17:"the insertion itself of the implant" is a generic term, can you specify this concept?
The assumption that "the implant insertion may create a damage of the implant connection" should be integrated with extra information, because implant companies
INTRODUCTION:
page 2, line 50: the term "implant-prosthetic index" is correct? I asked you again, as I checked on internet, and I found the term is "prosthetic index". However, I would suggest to use the same term in the manuscript and remove all the synonyms. As a suggestion, the most suitable term is "implant-abutment interface." Please read the reference below:
Zancopé K, Dias Resende CC, Castro CG, Salatti RC, Domingues das Neves F. Influence of the Prosthetic Index on Fracture Resistance of Morse Taper Dental Implants. Int J Oral Maxillofac Implants. 2017 Nov/Dec;32(6):1333-1337.
page 2, lines 62-63: "to the implant breakage or bone fractire" would be better than "breaking of implant or bone"
page 2, lines 64-65: You can remove the sentencd "Despite...connection"; I think that "however" is clear enough
page 2, lines 66-67: I would rephrase this sentence; as a suggestion:"Delgado et al. [21] showed that morphological changes occurred inside the internal connection when narrow implants were placed in dense bone; moreover, the deformation of the connection is accompanied with titanium particle release
page 2, line 70: "and for further developmentof a future ..." instead of "being this process favorable for the development and progression of a future peri-implant disease."
page 2, line 77: do you think that we can use the term "on the external connection" instead of "over the implant platform"?
MATERIALS AND METHODS:
can you specify that which implants have external/internal connection?
page 3, line 106: "indexation" instead of "index"
page 4, lines 133-134: you can specify here that the implants were inserted in "iuxtacrestal position" according to the manifacturer's protocol.
page 4, line 135: Ok, I get you point; therefore, you can say that: "the insertion torque of the implant was recorded", without saying that was 50-60 N
page 4, line 150: which software was used to obtain the 3D reconstruction? Please indicate name of the software, version, year, company.
page 4, line 152: you can specify "linear and angular measurements"
DISCUSSION:
I would remove lines 283-288, and I would move the findings of the study (lines 299-327). Try to summarize the results of your study, just to let the reader to catch them immediately.
page 12, lines 283-284: "The insertion torque was controlled for each... connections" should be moved in the results. In the discussion you can say: "In our study, we found that a 50-60 N of torque insertion resulted in morphological changes of the implant-abutment interface".
page 13, lines 321-323: please rephrase it; as a suggestion:" in implant type A, morphological alterations were located over the implant platform, where the implant carrier attaches to the implant. The inner part of the implant did not shox any morphological change".
page 13, line 326: which implant-abutment interface? Do you mean Morse taper implant-abutment connection?
page 13, line 326: I would say "may warrant long-term crestal bone stability, thanks to the mechanical properties of the conical connection" and I would remove "due to stable features and frictional stability of the conus. This concept has been evaluated, showing long-term crestal bone stability"
page 13, lines 331-332: I would rephrase this sentence. As a suggestion:" In order to overcome this limitation, wall thickness of the implant was analyzed.
page 13, lines 332-334: It is not clear, I would suggest to avoid long sentences.
page 14, line 340: please add a reference to this sentence. As a suggestion:
Patini R, Staderini E, Camodeca A, Guglielmi F, Gallenzi P. Case Reports in Pediatric Dentistry Journals: A Systematic Review about Their Effect on Impact Factor and Future Investigations. Dent J (Basel). 2019 Oct 24;7(4).
Author Response
Dear reviewer,
After reading some of your corrections, we perceive that there is something that has not been transmitted correctly, and it is necessary to clarify. Both types of implants, regarding the implant/abutment connection, are the same type: cone morse internal connection (there is no external connection), with the only difference that the type A implant does not have anti-rotational elements inside the connection; and type B has anti-rotational elements inside the connection.
What is really different in both implants evaluated is how the implant carrier hooks the implant to be inserted into the bone. In type A, four rectangular slots appear on the implant platform whose sole function is to serve as a support for the implant carrier to insert the implant. In type B, the system uses the anti-rotational elements located inside the connection to support the implant carrier. The justification for the present study precisely lies in this difference, in the way the implant is transported to be inserted. We defend that the anti-rotational elements located inside the connection itself should not be used to support the implant carrier, but rather elements location in a placement far from this point so critical for the posterior stability of the implant/prosthesis interface.
Having clarified this point, we proceed to the following changes as you recommended:
TITLE: I would suggest "Volumetric changes of different implant-abutment interfaces (rectangular slots vs hexagon connection) after placement in dense bone. In-Vitro Study" or "Volumetric changes of different implant-abutment interfaces (internal vs external connection) after placement in dense bone. In-Vitro Study"
We do not consider this change to be adequate, taking into account what was clarified in the previous paragraph, but we suggest change to: “Volumetric Changes in Morse Taper Connections After Implant Placement in Dense Bone. In-Vitro Study”
ABSTRACT:
page 1, lines 14-16: I would rephrase this sentence, e.g.: "The stability of the implant-abutment interface is crucial the maintenance of the implant integrity index". What is implant integrity index? Do you mean clinical long-term success?
The sentence has been changed as suggested. With “implant index integrity,” we mean the adequate morphology and volumetric relationship between the implant/prosthetic interface. Any change in both the implant connection or the prosthetic abutment would lead in a misfit who could end a biological or mechanical failure, so, yes, in some way, we also mean the long-term clinical success.
page 1, line 17: please change the term "structure" with "implant-abutment connection" and try to use this term for the whole article. As well, sometimes you use "conical connection" and sometomes "Morse taper connection"; the use of many synonyms is measliding for the readership. Please try to use the same term for the whole article "Morse taper implant-abutment connection".
The term has been changed for “morse taper implant/abutment connection” (lines 143, 347, 359, 362,… and so on).
page 1, line 17:"the insertion itself of the implant" is a generic term, can you specify this concept?
The sentence has been changed for “such as the actual insertion of the implant into the bone”
The assumption that "the implant insertion may create a damage of the implant connection" should be integrated with extra information, because implant companies
This extra information has been developed in the discussion and not in the abstract due to the few words allowed by the publisher (max. 200 words)
INTRODUCTION:
page 2, line 50: the term "implant-prosthetic index" is correct? I asked you again, as I checked on internet, and I found the term is "prosthetic index". However, I would suggest to use the same term in the manuscript and remove all the synonyms. As a suggestion, the most suitable term is "implant-abutment interface." Please read the reference below:
Zancopé K, Dias Resende CC, Castro CG, Salatti RC, Domingues das Neves F. Influence of the Prosthetic Index on Fracture Resistance of Morse Taper Dental Implants. Int J Oral Maxillofac Implants. 2017 Nov/Dec;32(6):1333-1337.
The implant abutment interface and the implant index are not exactly the same. The term “implant abutment interface” refers to the intimate relationship of the implant connection surface and the prosthetic abutment surface. On the other hand, the term “prosthetic index” refers to the anti-rotational components found inside the implant connection, which have their analogous elements in the prosthetic abutment to relate precisely and avoid the rotation of the prosthetic restoration.
An implant abutment interface may or may no contain a prosthetic index. Usually, there is always a prosthetic index contained inside the implant abutment interface in unitary restorations, but not always in multiple restorations
page 2, lines 62-63: "to the implant breakage or bone fractire" would be better than "breaking of implant or bone"
page 2, lines 64-65: You can remove the sentencd "Despite...connection"; I think that "however" is clear enough
page 2, lines 66-67: I would rephrase this sentence; as a suggestion:"Delgado et al. [21] showed that morphological changes occurred inside the internal connection when narrow implants were placed in dense bone; moreover, the deformation of the connection is accompanied with titanium particle release.
page 2, line 70: "and for further developmentof a future ..." instead of "being this process favorable for the development and progression of a future peri-implant disease."
Sentences changed as suggested.
page 2, line 77: do you think that we can use the term "on the external connection" instead of "over the implant platform"?
MATERIALS AND METHODS:
can you specify that which implants have external/internal connection?
As exposed at the beginning of the present comment-letter, the term “external connection” has been misunderstood. Both implants have the same morse taper internal connection and the elements over the implant platform (in type A implants) are part of the implant carrier engagement system.
page 3, line 106: "indexation" instead of "index"
page 4, lines 133-134: you can specify here that the implants were inserted in "iuxtacrestal position" according to the manifacturer's protocol.
page 4, line 135: Ok, I get you point; therefore, you can say that: "the insertion torque of the implant was recorded", without saying that was 50-60 N
Changed as suggested
page 4, line 150: which software was used to obtain the 3D reconstruction? Please indicate name of the software, version, year, company.
(SK-H Data Acquisition Software, version 1.0.3.0. (2014), Keyence Corporation, Osaka, Japan) included in line 150
page 4, line 152: you can specify "linear and angular measurements"
Of course, text modified.
DISCUSSION:
I would remove lines 283-288, and I would move the findings of the study (lines 299-327). Try to summarize the results of your study, just to let the reader to catch them immediately.
page 12, lines 283-284: "The insertion torque was controlled for each... connections" should be moved in the results. In the discussion you can say: "In our study, we found that a 50-60 N of torque insertion resulted in morphological changes of the implant-abutment interface".
page 13, lines 321-323: please rephrase it; as a suggestion:" in implant type A, morphological alterations were located over the implant platform, where the implant carrier attaches to the implant. The inner part of the implant did not shox any morphological change".
We have preceeded to make the changes as suggested
page 13, line 326: which implant-abutment interface? Do you mean Morse taper implant-abutment connection?
Yes, the term has been changed to avoid misunderstood.
page 13, line 326: I would say "may warrant long-term crestal bone stability, thanks to the mechanical properties of the conical connection" and I would remove "due to stable features and frictional stability of the conus. This concept has been evaluated, showing long-term crestal bone stability"
page 13, lines 331-332: I would rephrase this sentence. As a suggestion:" In order to overcome this limitation, wall thickness of the implant was analyzed.
page 13, lines 332-334: It is not clear, I would suggest to avoid long sentences.
Sentences changed as suggested.
page 14, line 340: please add a reference to this sentence. As a suggestion:
Patini R, Staderini E, Camodeca A, Guglielmi F, Gallenzi P. Case Reports in Pediatric Dentistry Journals: A Systematic Review about Their Effect on Impact Factor and Future Investigations. Dent J (Basel). 2019 Oct 24;7(4).
The recommended citation has been added.

Reviewer 2 Report
- Keywords must be presented in alphabetical order
-The authors do not discuss the limitations of the study, such as the reduced number of samples and the fact that they chose artificial bone
Author Response
Dear reviewer,
The following changes have been performed as you suggested:
- Keywords must be presented in alphabetical order
The order has been changed.
-The authors do not discuss the limitations of the study, such as the reduced number of samples and the fact that they chose artificial bone
The small number of samples is discussed in lines 392-397 and the use of artificial bone in lines 356-360.
